# Leveraging Institutional Support to Build an Integrated Multidisciplinary Care Model in Pediatric Inflammatory Bowel Disease

**DOI:** 10.3390/children8040286

**Published:** 2021-04-08

**Authors:** Jennifer Verrill Schurman, Craig A. Friesen

**Affiliations:** Division of Gastroenterology, Hepatology & Nutrition, Children’s Mercy Kansas City, 2401 Gillham Road, Kansas City, MO 64108, USA; cfriesen@cmh.edu

**Keywords:** pediatrics, inflammatory bowel disease (IBD), integrative care, integrated care, biopsychosocial, gastroenterology, psychology, interdisciplinary, multidisciplinary

## Abstract

While the biopsychosocial nature of inflammatory bowel disease (IBD) is now well accepted by clinicians, the need for integrated multidisciplinary care is not always clear to institutional administrators who serve as decision makers regarding resources provided to clinical programs. In this commentary, we draw on our own experience in building successful integrated care models within a division of pediatric gastroenterology (GI) to highlight key considerations in garnering initial approval, as well as methods to maintain institutional support over time. Specifically, we discuss the importance of making a strong case for the inclusion of a psychologist in pediatric IBD care, justifying an integrated model for delivering care, and addressing finances at the program level. Further, we review the benefit of collecting and reporting program data to support the existing literature and/or theoretical projections, demonstrate outcomes, and build alternative value streams recognized by the institution (e.g., academic, reputation) alongside the value to patients. Ultimately, success in garnering and maintaining institutional support necessitates moving from the theoretical to the practical, while continually framing discussion for a nonclinical/administrative audience. While the process can be time-consuming, ultimately it is worth the effort, enhancing the care experience for both patients and clinicians.

## 1. Introduction

There is an ever-increasing body of evidence supporting the biopsychosocial nature of inflammatory bowel disease (IBD). The biopsychosocial model recognizes that symptoms and disease activity result from multiple interacting pathways including biologic (e.g., physiologic, genetic), psychologic (e.g., emotional, behavioral) and social (relationships, environment) factors. It may appear obvious to clinicians that all of these areas represent therapeutic targets which need to be addressed to provide patients with optimal physical health and quality of life, and further that this dynamic model indicates a clear role for both the gastroenterology (GI) physician and the psychologist in partnership in routine IBD care. However, this need is not always so apparent to nonclinicians (e.g., administrators) who are often the decision makers regarding the resources provided to clinical programs. In our experience, administrators have two primary questions when considering support for the development of multidisciplinary and interdisciplinary clinics: (1) What are the patients’ needs and best practices for providing those needs? and, (2) What are the financial implications of the proposed practice model? With regard to obtaining support for the development of an IBD interdisciplinary program, the first question is more straight forward, more empirically supported, and more consistent from health system to health system. The second question addresses an area that continues to evolve and is somewhat dependent, at least initially, on the billing and contracting proficiencies of a particular health system. In this commentary, we will address both questions and describe methods we have found helpful for maintaining support after garnering initial approval. While the focus of this commentary will be on pediatric IBD, and the GI physician–psychologist partnership in IBD programs, the lessons learned can and should be applied to other pediatric subspecialty populations and clinical providers relevant to their care. A few examples of this broader application will be provided from our own lived experience.

## 2. Making the Case for Inclusion of a Psychologist in IBD Care

The first step in garnering support for embedding a psychologist in an IBD program, and creating a multidisciplinary clinic, is to use the existing literature to develop a compelling story for inclusion of a psychologist in routine IBD care. The following is not meant to be a comprehensive review of the literature but rather one example of how to use the current available literature to effectively highlight the importance of mental health in IBD to an administrative leader. The main points to cover include the fact that psychological dysfunction is common in IBD, impacts the clinical course affecting symptoms, quality of life, and resource utilization, and is amenable to treatment which, in turn, has a positive impact on the disease course and costs.

Sample rationale: Pediatric patients with IBD are at increased risk for depression, anxiety, social isolation, and altered self-image [1,2]. In fact, nearly 25% of adolescents with IBD have symptoms of depression, and most go unrecognized without intentional screening and evaluation [3]. Internalizing symptoms, in turn, are associated with increased pain frequency and severity and greater pain impact in pediatric IBD [4,5], as well as lower health-related quality of life [6]. In adolescents with IBD, depression correlates with pain, diarrhea, and weight loss [7]. More broadly, stress is associated with symptom exacerbation and relapse, particularly in patients with anxiety and depression [8]. Given all of this, it is perhaps unsurprising that a mental health diagnosis significantly increases emergency department visits and inpatient stays in youth with IBD [9]. A recent systematic review further demonstrated that anxiety and depression were negatively correlated with transition self-efficacy, likely delaying and/or disrupting the successful transfer to adult care [10]. The negative impact that psychologic dysfunction can have on treatment adherence appears key in understanding the relationship between internalizing symptoms and IBD control. Adherence is negatively affected by several barriers, including (but not limited to) the complexity of the medical regimen in pediatric IBD [11,12]. Emotional and behavioral functioning contribute significantly to nonadherence [13] and appear to moderate the relationship between adherence barriers and adherence, with the lowest adherence seen in patients with higher barriers in association with higher anxiety and/or depression [14,15]. Nonadherence rates of 2 to 93% have been reported in youth with IBD, depending on measurement method, and is associated with poor coping strategies, anxiety, depression, and difficulties with family and social interactions [16]. In turn, nonadherence is associated with increased disease severity, lower rates of clinical remission, increased risk of relapse, and increased healthcare costs [13,17,18]. In sum, anxiety and/or depression are now known to be common in pediatric IBD, and to negatively affect IBD symptoms and disease course, adherence, quality of life, and transition efficacy/readiness.

Fortunately, strategies exist for identifying and effectively treating at risk patients with IBD. Appropriate screening and treatment of anxiety and depression in youth with IBD can positively impact the clinical course and improve health-related quality of life [7]. Psychotherapy in children and adolescents with IBD and depression improves depression and general adjustment, lessens impairment, and reduces healthcare utilization including hospital admissions, emergency department visits, radiologic exams, and endoscopies [19,20]. Likewise, individually tailored psychological treatment of nonadherence, including training in problem solving skills and—in some cases—behavior management, self-management, and education, can significantly improve adherence rates [21,22,23,24]. Given this mounting evidence, both the North American Society for Pediatric Gastroenterology, Hepatology, and Nutrition (NASPGHAN) and a task force of the ImproveCareNow network have endorsed the importance of routine psychosocial and adherence screening in youth with IBD, marking the new standard for best practice and setting the stage for psychological treatment to be integrated routinely into clinical care [2,13,21].

While it is important to share the rich body of evidence that supports the need for a psychologist as an integral member of the healthcare team, as above, our experience has been that collecting data to bring this literature closer to home is often vitally important. Data collected from your own clinic population can make the need for more holistic care feel more personal. We would recommend instituting local screening even before seeking approval to expand the healthcare team to include psychology (or any other discipline). Ultimately, the choice of specific screening tools must be driven by a variety of considerations: (1) the topic areas to be screened; (2) how often these areas are to be reassessed; (3) the availability of measures with solid psychometric properties within those areas; 3) the length of specific measures being considered relative to the length of the entire battery; (4) the cost of using specific measures as part of standard clinical care; (5) the logistics of administering and scoring specific measures; and, last but certainly not least, (6) the clinical utility of each measure in driving treatment decisions. Based on the previous rationale provided, the local clinic population for pediatric IBD should be screened, at a minimum, for depression and anxiety, as well as for adherence and adherence barriers. While it is beyond the scope of this commentary to review the array of (and gaps in) available screening tools, Figure 1 provides an example of the screening measures currently in use within our own IBD program as part of the assessment of newly diagnosed patients, as well as where/how these measures fit into the flow of our integrated care model. Figure 2 provides the same for our clinic designed for young adults with IBD working toward transition and transfer from pediatric to adult healthcare.

Other information is also likely to be available in the electronic medical record (EMR) that can be abstracted and combined with this screening data to help you make your case. For pediatric IBD, the effects of anxiety, depression, and adherence could be assessed for their relationship to EMR-based outcome measures including, for example, steroid-free admission rate, emergency department visits, and hospital admission and length of stay at the population level. The more that you can document the impact of psychologic and social factors on IBD symptoms, disease course, and related costs within your own clinic population, the easier you will find it to “ask” for the inclusion of a psychologist in routine IBD care.

Finally, we recommend emphasizing that when we take on the responsibility to care for a patient with IBD, we also accept the responsibility to identify and treat all relevant factors which can affect outcome. In short, once you screen patients, there is an implied ethical obligation to address the issues identified. The existing literature, hopefully amplified by local data, provides a strong case that mental health screening and treatment should be an integral component of an IBD program.

## 3. Justifying an Integrated Model for Delivering Care

Once an administrator is convinced of the necessity of having a psychologist as part of the IBD healthcare team to ensure optimum disease outcomes, it still leaves open the question of what is the most efficient, cost-effective, and patient-centered approach for involving the psychologist. For example, why not screen in clinic and refer patients to a psychologist as needed? What are the advantages of a multidisciplinary program where each professional contributes to the care of the patient or an interdisciplinary program where professionals also directly interact with shared decisions taking into account the numerous factors contributing to outcomes? Various approaches have been used to determine the best care models from both a healthcare provider and a patient perspective and they unanimously support a “multidisciplinary” process, which in reality often describes an integrated multidisciplinary or interdisciplinary approach to practice [32,33,34,35]; see [36] for an in-depth discussion of various multidisciplinary care models. In a study utilizing semistructured interviews with IBD specialists, the consensus was that the ideal model involved “sharing collective expertise in a formalized manner” with interactions between subspecialty providers (i.e., GI physician and psychologist) being of the highest importance [32]. A systematic review of IBD care standards from both a healthcare provider and patient perspective also endorsed an interdisciplinary coordinated structure and concluded that this model was cost effective [33]. In the UK, healthcare providers and patients achieved near unanimous consensus that care should be provided by a multidisciplinary team which meets regularly to discuss appropriate patients [34]. Another recent systematic review also concluded that an integrated care model incorporating a healthcare team that included a psychologist was the optimum model and is shown to decrease hospital admissions, IBD surgeries, comorbidities, and both direct and indirect costs [35]. It is clear that a care model that falls along the care spectrum between integrated multidisciplinary (in which providers work side-by-side with real-time meaningful communication/collaboration) and interdisciplinary (in which providers routinely see patients together) is supported by both providers and patients.

In addition to sharing empiric evidence in favor of integrated multidisciplinary/interdisciplinary care models (hereafter referred to as integrated care), we have found it helpful to describe personal observations to support why an integrated approach is preferred by providers and patients alike. Providers involved in the care of IBD patients recognize the interaction of various factors that determine patient outcomes across the biopsychosocial model and the value of discussion in real-time with shared input to create the optimum care plan. For example, nonadherence is a significant factor in outcome as medications cannot be effective if they are not consumed. It is therefore important to assess adherence and adherence barriers in the clinic, as well as psychosocial factors which may be driving nonadherence. This allows nonadherence to be addressed in clinic and decisions to be made regarding treatment. Is the barrier that the medication must be taken at a frequency that does not work with the patient’s schedule and commitments? If so, thoughtful consideration can be given to changing the medication or dosing regimen with the GI physician present. Is anxiety or depression the primary barrier? If so, a treatment plan to address internalizing issues can be laid out quickly with a psychologist present. From a patient perspective, identifying a psychosocial issue or nonadherence which prompts a referral to a psychologist at a future date, who in turn communicates back with the medical provider who subsequently makes medication decisions, is an unsatisfactory experience and a needless delay in care [37,38]. The patient experience is enhanced by being able to see the relevant providers in a single visit and know that their care team is aligned in any treatment recommendations.

## 4. Addressing Program Finances

Addressing the financial impact of a program can be the most challenging component of selling your integrated care vision. Approximately half of IBD healthcare providers perceive funding to be the greatest barrier to implementing the ideal care model [35]. The challenges include the wide variety of evolving reimbursement models, trends towards capitated care and shared risk models, and possible future reimbursement models centered around delivering value to the patient, i.e., improved outcomes at decreased costs. While pay-for-service is still the primary payment mechanism, healthcare providers need to plan for the movement to payment systems that value high-quality care [39]. While a discussion of the varying and evolving payment models is beyond the scope of the current paper, we would emphasize three points in preparation for discussion of the program proposal with administration: (1) Develop a detailed plan of the care model; (2) Develop a business plan; and, (3) Emphasize that finances need to be assessed as total inflow minus total outflow.

First, a detailed description of the ideal clinical care model allows administration to understand personnel and resource needs, as well as plans to maximize efficiency. In addition, it provides the solid framework needed for development of a business plan, as there can be more accurate assessment of start-up funds needed and on-going costs. The business plan should also include any expansion into new markets, projected revenue, and a marketing plan. Greater detail in both the clinical care model and business plan tends to breed greater confidence within administrative decision-makers. It cannot be over-emphasized that business partners need to be identified to provide data credible to administration. However, it is equally important that the clinical team remains closely involved in business planning to prevent data being derived from faulty assumptions and, instead, ensuring that they are accurate based on the experience and plans of the clinical team. Collaboration on the business plan allows for the creation of the most accurate, meaningful, and credible projections. Probably the most important factor in selling and sustaining an integrated care vision is to frame the financial assessment of the ideal clinical care model as total revenue minus total expenses for the program as a whole, diverting the discussion away from the analysis of individual team members. Given the different billing parameters, reimbursement rates, etc. between subspecialty providers, the close examination of individual team members is likely to dismantle an integrated care program. While optimizing billing for individual team members will help the bottom line, looking at the whole (i.e., collective inflow versus outflow) more effectively takes into account time spent by the team, especially when one team member can bill for a portion of the clinical service (e.g., time spent in non-face-to-face interdisciplinary collaboration) and another cannot. If the whole program meets the financial thresholds set by the administration, it should be approved in total. If not, then parameters can be defined and efforts made to reconcile finances without compromising the highest quality of care. As more reimbursement becomes capitated or based on the quality of outcomes, integrated care models are well suited to be the most cost-effective, demonstrating improved outcomes that in turn lower costs and improve patient value.

## 5. On-Going Support

While existing literature supports the need for a psychologist, the superiority of integrated care, and the financial benefits, this body of evidence only provides projections and theoretical benefits in the eyes of administration. It is vitally important to confirm the accuracy of these projections in actual practice once the program is implemented. This is best accomplished by systematic data collection and utilization of a quality improvement perspective to demonstrate need and value. Ideally, the data would be compared over time, be used to guide care, and be published. This demonstrates professional commitment and provides marketing opportunities to enhance program recognition and reputation, which also serves the larger institution. Publishing data related to psychological factors and adherence provides credibility to the initial justification and emphasizes that these are issues in the actual population being served in a program—both critical to sustaining an integrated care model long-term.

An important practical consideration in publishing on clinical data is the Institutional Review Board (IRB) oversight and appropriate timing for review. The right timing needs to be decided before any data is collected and depends on the scope of what you plan to collect, whether you already have specific research questions in mind, and the type of project you want to pursue. In terms of scope, if you can justify each and every measure in your screening battery as impacting clinical care and treatment decision making for the individual patient, and are including the results of all screening measures in your clinical documentation, then you can consider this clinical data capture and not research. In this case, no IRB is required up front. Instead, if/when you develop a research question that you think you can answer with the clinical data on hand, you would submit a retrospective chart review protocol to your local IRB for review at that time. In this scenario, your clinical data is simply another type of EMR data that you can abstract within an IRB-approved retrospective chart review. However, if there is even one component of your screening battery that you cannot justify form an immediate clinical perspective, then things get a bit more complicated. One option is to separate that measure (or measures) from the rest of the clinical screening battery and treat that measure as research. In this case, you would submit an IRB protocol up front to allow you to ask families to complete this “study” measure and grant permission to allow you to also collect specific information from the EMR (including the clinical screeners, if desired). This is a relatively clean approach in that families would still complete all other screeners as standard clinical care and only complete the “study” measure if they consent to participate in your IRB-approved research study. Another option is to submit your screening battery to the IRB with a request to create a repository, which allows you to ask permission of families up front to use all data collected as part of the clinical program as to answer relevant research questions as they arise. This situation requires more up front work on approval, but it can simplify your research process on the back end as you do not need to go back to the IRB with repeated requests for retrospective chart review later on. Further, getting approved for a repository opens up the possibility of including measures that you might not have included in a purely clinical database. However, you need to ensure that you have an error-proof way of separating those patients whose families have consented to have their data included in the repository from those who have not, even though they are completing the same (or at least a similar) battery as part of clinical care if they do not consent. This is critical to prevent accidentally including families in your research who have actively opted out at an earlier timepoint. Finally, some projects may fall into the quality improvement (QI) realm, and be exempt from IRB review. Such projects typically focus on closing a known gap in care, such as improving vaccination rates in IBD patients, increasing the percent of patients seen for a clinic visit within the recommended follow up period, or even improving the rates of behavioral health follow up in patients who screened above a depression cutoff. The process for QI determination differs from institution to institution, but specific guidance should be available by calling your local IRB office.

Examples of work published from the IBD program at our institution include, among others, demonstration of the value of screening for psychological dysfunction [40], examination of patient and family perspectives on barriers to transition and transfer of care [41], documentation of high rates of both accidental and volitional nonadherence associated with increased disease severity and decreased quality of life [42], and development of a measure to assess accuracy of patient and caregiver knowledge of disease and treatment regimen [43]. Although pediatric IBD is the focus of this commentary, we have employed a similar approach across several other chronic GI conditions with success. Perhaps most notably, and certainly the most long-standing of which is our chronic abdominal pain program. With this program, we have been able to improve our clinical data collection iteratively over time to refine our process, align them with emerging knowledge, generate new knowledge, and utilize our data for individual patient monitoring and programmatic quality improvement efforts. Specifically, we have published multiple papers demonstrating psychologic disturbances in patients with chronic abdominal pain including interactions with functional disability, sleep, pain severity, and gastrointestinal inflammation [44,45,46,47]. We have published outcome data for a large population of children with chronic abdominal pain treated in our interdisciplinary program [48]. Further, we have demonstrated that anxiety and sleep disturbances are associated with poorer initial response to treatment in youth with chronic abdominal pain [48]. In one of our newer programs, we have demonstrated that poor adherence to lab testing in a liver transplant population is associated with significant consequences [49]. These manuscripts, like those noted in IBD above, have provided tangible peer-reviewed data demonstrating an on-going need for psychology as an integral part of the care program. Of note, it has been helpful to collect data on patient and family satisfaction and outcomes. We were able to demonstrate that an integrated care model for chronic abdominal pain that involved a GI physician and a psychologist was associated with increased caregiver satisfaction and greater receptivity to adhere to treatment recommendations (both medical and psychosocial) [50], thus providing indirect support for all of our integrated care programs. Most of these examples were made possible by systematically collecting data across biologic, psychologic, and social realms with subsequent evaluation of the interactions between these factors and with outcomes.

Patient-reported outcomes and satisfaction, along with the incidence of comorbidities, emergency department visits, and hospitalizations, should be combined with financial performance metrics to determine value, particularly from a patient perspective. We would recommend reporting this data up the administrative ladder on a regular basis (e.g., semiannually). The preemptive sharing of data, particularly when there are tangible academic products included, tends to decrease the chances of the program being part of future administrative discussions regarding budget cuts. While the most important goal is to build an excellent program which delivers outstanding patient care and outcomes, program maintenance and growth can be enhanced by taking every available opportunity to demonstrate the importance and value to the financial decision makers.

## 6. Conclusions

For clinicians, the discussion has moved well past whether psychologists should be considered an integral member of the IBD healthcare team to how to best implement integrated care in practice. Our goal in this commentary was to describe methods we have found helpful to leverage institutional support to build and maintain integrated multidisciplinary care models in pediatric IBD and other GI populations, but these steps and lessons learned could easily be applied to the building of integrated programs in other pediatric subspecialties, as well. First and foremost, to attain the necessary resources to optimize care and outcomes, the healthcare team needs to undertake an intentional process to educate administrative leaders on the current literature, local patient needs, and best practices. It also is imperative to directly assess program finances (i.e., collective inflow versus outflow) and value delivery in both the initial process and on an on-going basis. While we have offered some advice, it is essential to make an assessment that incorporates current institutional billing capabilities, while also identifying opportunities for improving billing practices. Finally, success in garnering and maintaining institutional support necessitates moving from the theoretical to the practical, by collecting and using local data to demonstrate need, outcomes, and value to the institution. While the process can be time-consuming, ultimately it is worth the effort, enhancing the care experience for both patients and clinicians.

## Figures and Tables

**Figure 1 children-08-00286-f001:**
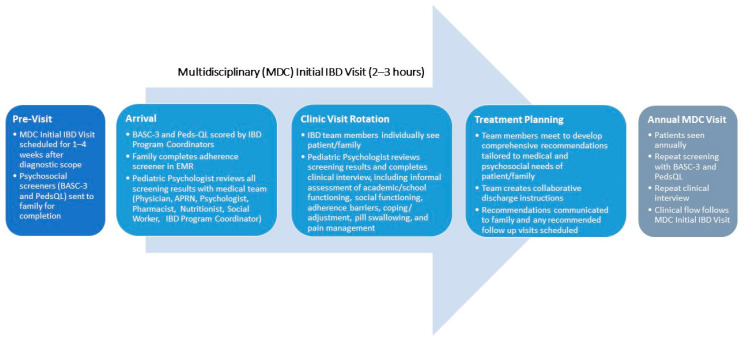
Sample flow diagram with inclusion of screening process for new patient integrated care clinic for pediatric inflammatory bowel disease (IBD). MDC: multidisciplinary clinic; APRN: advanced practice nurse; EMR: electronic medical record. Screening measures referenced above: BASC-3, Behavioral Assessment System for Children–Third Edition [25]; Peds-QL, Pediatric Quality of Life Inventory 4.0 Generic Core Scales [26]. For more information on this clinic and the screening process/questions used, please see Maddux and colleagues [27].

**Figure 2 children-08-00286-f002:**
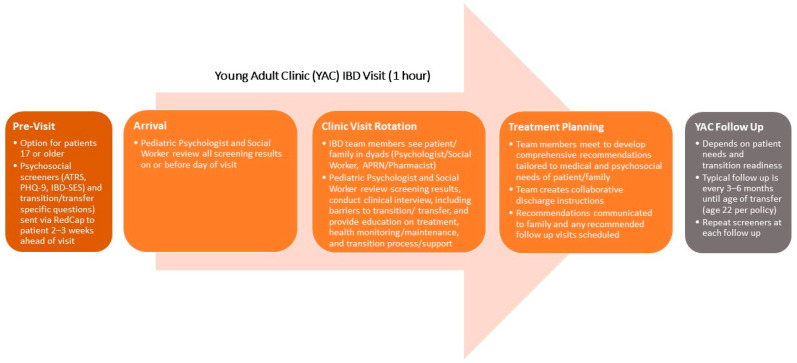
Sample flow diagram with inclusion of screening process for young adult integrated care clinic for pediatric IBD. Screening measures referenced above: ATR, Allocation of Treatment Responsibility Scale [28]; PHQ-9, Patient Health Questionnaire-9 [29]; IBD-SES, IBD Self-Efficacy Scale [30]. For more information on this clinic and the screening process/questions used, please see Maddux and colleagues [31].

## Data Availability

Not applicable.

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
