# Peer review of "Leveraging Institutional Support to Build an Integrated Multidisciplinary Care Model in Pediatric Inflammatory Bowel Disease"

_children, 2021, doi:10.3390/children8040286_

Round 1

Reviewer 1 Report

This narrative review was informative.
1. Psychological approach in the care of IBD patients is essential not only in children but also in adults. Is there a difference in approach methods between children and adults? If so, what are they? If the approach is complex in children, what are the means to overcome these difficulties?
2. Assessing psychosocial factors contributing to non-adherence is very important to improve the clinical outcome of IBD. If possible, it would be easier for readers to understand if you show specific intervention methods in a Table or other format. 
3. This reviewer understands that psychologists' intervention is essential, but what kind of approach is needed in the future to promote active intervention? This reviewer would like to ask the authors to discuss how to approach this issue, referring to the literature to date with creating figures for making the readers understand easily.

Author Response

Point 1. Psychological approach in the care of IBD patients is essential not only in children but also in adults. Is there a difference in approach methods between children and adults? If so, what are they? If the approach is complex in children, what are the means to overcome these difficulties?

Response 1: We appreciate the reviewer’s question about the difference between children and adults, and desire to have us include more information on application to adult care. However, we feel strongly that this is outside the scope of both the current commentary and the journal itself, both of which focus on children.

Point 2. Assessing psychosocial factors contributing to non-adherence is very important to improve the clinical outcome of IBD. If possible, it would be easier for readers to understand if you show specific intervention methods in a Table or other format.

Response 2: We now include two examples of integrated care clinics within our pediatric IBD program (Figures) that we hope serve to clarify how screening (with specific examples) fits into clinic flow and informs treatment. Further discussion of specific intervention methods is outside the scope of the current commentary, but can easily be found in several of our references.

Point 3. This reviewer understands that psychologists' intervention is essential, but what kind of approach is needed in the future to promote active intervention? This reviewer would like to ask the authors to discuss how to approach this issue, referring to the literature to date with creating figures for making the readers understand easily.

Response 3: Please see comment above.

Reviewer 2 Report

Thank you for the opportunity to review this commentary, which outlines practical steps to take in building an integrated, multidisciplinary model for the treatment of pediatric Inflammatory Bowel Disease (IBD) within a division of gastroenterology. This manuscript builds on existing literature that supports the clear benefits of a multidisciplinary care model and offers a “how to” for readers, that I believe could also be applied to other pediatric specialty care clinics/diseases, making this a clinically meaningful and potentially very impactful paper.

A few comments and suggestions are offered to maximize clarity and perhaps even make this piece more broadly applicable (i.e., across pediatric specialties).

Abstract:

  • Page 1, Line 12: consider changing “IBD programs” to “clinical programs” to highlight this is a scenario and a potential struggle for all clinical programs and is not specific to IBD programs.
  • Page 1, Line 19: I believe the authors meant “alternative.”

Intro:

  • Page 1, Line 37: as above, consider broadening “IBD programs” to “clinical programs” to increase applicability to a broader range of readers.
  • Page 2, Line 49/50: Similarly, consider broadening “other GI populations” to “other pediatric specialty programs.”

Section 1:

  • Page 2, Line 73: I believe “visit” should be “visits.”
  • Page 2, Line 92: Can you clarify “poor functioning”?
  • Page 2, Line 95: Related, can you clarify “functioning”?
  • Page 3, Line 105: consider some kind of break or other indication that the sample literature review/rationale has ended.
  • Page 3, Line 108: I’m not sure the authors’ intended meaning is coming through clearly. I recommend rewriting the sentence that begins, “Local data can…”
  • Page 3, Line 108-110: The sentence that begins “We would recommend” could use some clarification…
  • Might the authors consider included some recommended measures for screening anxiety, depression and barriers to adherence? Or for assessing adherence?

Section 2:

  • Page 3, Line 133: “have” should be “has”
  • Page 3, Line 134: “patient’s” should be “patient”
  • Page 4, Line 160: are any medications effective if not consumed?
  • Page 4, Line 173: I believe the first “in” should be “is.”
  • Again, I believe this paper could be even more helpful to readers if specific adherence and psychological screening measures were included- perhaps the measures this particular program is utilizing?

Section 3:

  • Page 4, Line 177: consider explicitly describing “selling your integrated care vision to hospital administration.”

Section 4:

  • Might the authors consider including brief discussion or at least mention of IRB-approval and/or QI-designation process or considerations related to the clinical data collected and published?
  • Have you used any specific treatment satisfaction/acceptability measures that have worked well in your setting that might be helpful to share with the readers?
  • Page 6, Line 261: Do the authors have specific advice on what domains of patient-reported outcomes might be most meaningful to collect? Or specific measures that have worked well in their setting?

Section 5:

  • I suggest including a comment that these steps and the insight shared here might be applicable to providers working to build integrated programs in other pediatric specialty care areas.

General comments:

  • The manuscript should be carefully edited to correct a few grammatical and mechanical mistakes (such as those outlined above) and to improve overall readability. Here are several sentences I believe could be strengthened with some editing:
  • Page 2, Line 58: “While this information…”
  • Page 2, Line 66: “While these mental health…”
  • Page 3, Line 143: “A process was undertaken…”
  • Page 4, Line 189: “First, a detailed…”
  • Page 5, Line 253: “Again, within our…”
  • Page 6, Line 261: “At a minimum,…”

Author Response

Response to Reviewer 2 Comments

Thank you for the opportunity to review this commentary, which outlines practical steps to take in building an integrated, multidisciplinary model for the treatment of pediatric Inflammatory Bowel Disease (IBD) within a division of gastroenterology. This manuscript builds on existing literature that supports the clear benefits of a multidisciplinary care model and offers a “how to” for readers, that I believe could also be applied to other pediatric specialty care clinics/diseases, making this a clinically meaningful and potentially very impactful paper.

A few comments and suggestions are offered to maximize clarity and perhaps even make this piece more broadly applicable (i.e., across pediatric specialties).

Point 1. Abstract:

Page 1, Line 12: consider changing “IBD programs” to “clinical programs” to highlight this is a scenario and a potential struggle for all clinical programs and is not specific to IBD programs.

Page 1, Line 19: I believe the authors meant “alternative.”

Response 1. Thank you, all changes have been made as suggested.

Point 2. Intro:

Page 1, Line 37: as above, consider broadening “IBD programs” to “clinical programs” to increase applicability to a broader range of readers.

Page 2, Line 49/50: Similarly, consider broadening “other GI populations” to “other pediatric specialty programs.”

Response 2. Thank you, all changes have been made as suggested.

Point 3. Section 1:

Page 2, Line 73: I believe “visit” should be “visits.”

Page 2, Line 92: Can you clarify “poor functioning”? Rephrased to be more specific.

Page 2, Line 95: Related, can you clarify “functioning”? Rephrased to be more specific.

Page 3, Line 105: consider some kind of break or other indication that the sample literature review/rationale has ended. Added title and indentations to make offset clear.

Page 3, Line 108: I’m not sure the authors’ intended meaning is coming through clearly. I recommend rewriting the sentence that begins, “Local data can…”

Page 3, Line 108-110: The sentence that begins “We would recommend” could use some clarification…

Might the authors consider included some recommended measures for screening anxiety, depression and barriers to adherence? Or for assessing adherence? While a review of available screening measures is well beyond the scope of this commentary, we added guidance for selection and examples currently in use in our two new Figures depicting clinic flow through two integrated care clinics within our IBD Program.

Response 3. All other changes have been made as suggested if not specifically addressed above.

Point 4. Section 2:

Page 3, Line 133: “have” should be “has”

Page 3, Line 134: “patient’s” should be “patient”

Page 4, Line 160: are any medications effective if not consumed?

Page 4, Line 173: I believe the first “in” should be “is.”

Again, I believe this paper could be even more helpful to readers if specific adherence and psychological screening measures were included- perhaps the measures this particular program is utilizing? See comment in previous section.

Response 4. All other changes have been made as suggested if not specifically addressed above.

Point 5. Section 3:

Page 4, Line 177: consider explicitly describing “selling your integrated care vision to hospital administration.”

Response 5. We found this comment a bit confusing, as the entire paper is really about this very issue. That said, we would be happy to include further details if there is something specific that this reviewer feels is still missing from the commentary.

Point 6. Section 4:

Might the authors consider including brief discussion or at least mention of IRB-approval and/or QI-designation process or considerations related to the clinical data collected and published? We have added a section on this issue.

Have you used any specific treatment satisfaction/acceptability measures that have worked well in your setting that might be helpful to share with the readers? Please see comment above about the handling of screener selection and provision of limited examples.

Page 6, Line 261: Do the authors have specific advice on what domains of patient-reported outcomes might be most meaningful to collect? Or specific measures that have worked well in their setting? Examples have now been provided both in the text and in the Figures.

Response 6. Please see specific responses above.

Point 7. Section 5:

I suggest including a comment that these steps and the insight shared here might be applicable to providers working to build integrated programs in other pediatric specialty care areas.

Response 7. Done.

Point 8. General comments:

The manuscript should be carefully edited to correct a few grammatical and mechanical mistakes (such as those outlined above) and to improve overall readability. Here are several sentences I believe could be strengthened with some editing:

Page 2, Line 58: “While this information…”

Page 2, Line 66: “While these mental health…”

Page 3, Line 143: “A process was undertaken…”

Page 4, Line 189: “First, a detailed…”

Page 5, Line 253: “Again, within our…”

Page 6, Line 261: “At a minimum,…”

Response 8. All changes have been made as suggested.